# Improved Catch Fish Optimization Algorithm with Personalized Fishing Strategy for Global Optimization

1st Bowen Xue
*School of Electrical and Information Engineering*
*Northeast Petroleum University*
Daqing, China
xuebowen@stu.nepu.edu.cn

2nd Heming Jia*
*School of Information Engineering*
*Sanming University*
Sanming, China
jiaheming@fjsmu.edu.cn

3rd Honghua Rao
*School of Electrical and Information Engineering*
*Northeast Petroleum University*
Daqing, China
20200862235@fjsmu.edu.cn

4th Jinrui Zhang
*School of Information Engineering*
*Sanming University*
Sanming, China
ruiruiz2308@163.com

5th Yilong Du
*School of Information and Electrical Engineering*
*Heilongjiang Bayi Agricultural University*
Daqing, China
wy15093488812@163.com

6th Zekai Ai
*College of Design and Engineering*
*National University of Singapore*
Singapore
aizekai@u.nus.edu

*Abstract*—Catch Fish Optimization Algorithm (CFOA) is a new meta-heuristic optimization algorithm with human behavior. In this algorithm, search agents simulate the process of rural fishermen fishing in the pond. Therefore, the CFOA generally consists of two phases of the update: the exploration phase and the exploitation phase. However, it still falls under the local optimum and has a low convergence rate. To this end, we propose an improved catch fish optimization algorithm(ICFOA) based on personalized fishing strategies. First, the adaptive Gaussian perturbation is adopted to the exploration stage process to increase the global search capability, expand the search range, and improve efficiency while avoiding falling into the local optima. Then, based on the personalized fishing strategy, the personal position of fishermen is updated by randomly selecting "freehand fishing" factors or "using fishing net" factors to accelerate the algorithm's convergence speed. Furthermore, comparative experiments were performed using the CEC2020 test suite to compare the performance of ICFOA and other excellent meta-heuristics. Finally, Wilcoxon's rank-sum test was used to verify the validity of our statistical experimental results. Moreover, the performance of ICFOA in reducer design also indicates that ICFOA can get the optimal solution in solving practical engineering optimization problems. The results show that ICFOA has a more competitive performance than the original CFOA.

*Keywords—Catch Fish Optimization Algorithm，adaptive Gaussian perturbation, Personalized Fishing Strategy*

## I. INTRODUCTION

In the current era of rapid technological advancement, optimization problems hold a critical position across various domains, including engineering design, economic management, and computer science. Examples include the Crayfish Optimization Algorithm (COA) [1], Whale Optimization Algorithm (WOA) [2], and Grey Wolf Optimization (GWO) [3]. COA excels in exploration but may converge slowly. WOA balances exploration and exploitation well but can get trapped in local optima. GWO is strong in convergence but requires careful parameter tuning.

Human behavior-based optimization algorithms are a class of optimization techniques designed to tackle complex optimization problems by emulating human or other biological behaviors and decision-making processes. By mimicking natural phenomena such as evolution, foraging, and social interactions, these algorithms can effectively search and optimize complex solution spaces.

In solving problems related to economic scheduling, functional optimization, and engineering design, human behavior-based optimization algorithms are especially adept at avoiding local optima and discovering global optima or solutions close to the global optimum. For instance, the Human Behavior-Based Optimization (HBBO) [4] algorithm models human behavior patterns, particularly focusing on how humans learn and solve problems through interaction and communication. This algorithm integrates multiple human behavioral traits, such as experiential learning, imitation, social interaction, and collaboration, to achieve efficient search and optimization in complex problems.

Heming Jia et al. [5] In 2024, an innovative optimization algorithm inspired by human behavior, namely the catch fish optimization algorithm(CFOA). The main inspiration for the CFOA comes from the fishing practices of the fishermen. The CFOA contains updated rules based on different fishing practices. As intelligent humans, fishermen often use a variety of ways to find fish, such as sharing fishing experiences, using different fishing tools, etc., so their location update rules are based on individuals and teams. Furthermore, as capture rates decline, fishermen will choose whether to change their fishing strategy. Experimental results show that the proposed algorithm outperforms others in finding the optimal solution and convergence speed. However, as stated by the NFL Theorem [6],given the diversity and complexity of optimization problems, no universal algorithm can be directly applied to address all types of optimization challenges. This reality requires the exploration and adoption of more rigorous and targeted strategies to continuously improve and optimize the algorithm design.

Just as the problems faced by many optimization algorithms, the original CFOA algorithm found it difficult to completely avoid the limitations of low convergence

* Corresponding author.
This work is supported by the Natural Science Foundation of Fujian Province under Grant 2021J011128.

efficiency and easily fall into local optimal solutions in specific optimization tasks. Given this, the optimization and upgrading of the CFOA algorithm can not only improve the efficiency of its algorithm but also broaden its scope of application. Therefore, this paper presents an improved CFOA algorithm (ICFOA) based on a personalized fishing strategy. PFS greatly enhances the solving performance of CFOA in complex optimization problems. At the same time, the position of the fishermen is updated based on the personalized fishing strategy, which not only makes the algorithm more detailed and comprehensive when searching for the solution space but also enhances the algorithm's ability to escape local optima and find the global optimal solution. Finally, to test ICFOA, to test the improved optimization algorithm, this paper utilizes ten commonly used benchmark functions and chooses the optimization algorithm (CFOA) for evaluation. and five representative meta-heuristic algorithms for comparative experiments to validate the effectiveness and advantages of ICFOA.

The remainder of this paper is structured as follows: Section II provides the concept of the original CFOA, Section III details the proposed algorithm ICFOA, Sections IV and V demonstrate the experiment analysis in comparison with several popular metaheuristics under the CEC2020 test suite and Wilcoxon's rank-sum test, and Section VI concludes.

## II. Catch Fish Optimization Algorithm

The CFOA simulates the fishing behavior of village fishermen. To catch fish more easily, fishermen choose different fishing methods to catch fish. Similar to other metaheuristic algorithms (MAs), CFOA consists of three distinct stages: initialization, exploration, and exploitation.

### 1) Initialization phase

The matrix $F$ represents the location data of $N$ search agents in a $d$-dimensional space, and the formula is shown below:

$$F = \begin{bmatrix} F_{1,1} & F_{1,2} & \cdots & F_{1,n} \\ F_{2,1} & F_{2,2} & \cdots & F_{2,n} \\ \vdots & \vdots & \ddots & \vdots \\ F_{n,1} & F_{n,2} & \cdots & F_{n,n} \end{bmatrix}_{N \times d} \quad (1)$$

$$F_{i,j} = lb_j + \left(ub_j - lb_j\right) \times \text{rand} \quad (2)$$

The matrix F represents the position information of N search agents within a d-dimensional space. Its initialization formula is as follows: Fi,j denotes the position of the ith agent in the jth dimension, where ubj and lbj represent the maximum and minimum limits of the jth dimension, respectively, rand is a random number in the interval (0,1).

Using the current position data of each fisherman, we apply the fitness evaluation function *fobj* to determine their fitness scores, yielding the following fitness matrix:

$$f = fobj(F) = \begin{bmatrix} f_1 \\ f_2 \\ \vdots \\ f_N \end{bmatrix} \quad (3)$$

In the above formula, $f_1$ represents the fitness value of the first fisherman, $f_2$ denotes the fitness value of the second fisherman, and so on. We use a value of 0.5 to evenly distribute the balance between exploitation and exploration across iterations. In the initial part of the phase (when EFs/MaxEFs $<$ 0.5), individuals focus on global exploration, while during the latter part of the phase (when EFs/MaxEFs $\geq$ 0.5), they shift towards exploitation.

### 2) Individual and group fishing (exploration phase)

When fishermen explore, initially mainly through independent search and using group encirclement as an aid. As the exploration proceeds, the environmental advantages gradually shift from the fish side to the fishermen. In addition, continuous capture will lead to a decrease in fish population and capture rate. Fishermen will shift from independent exploration to mainly relying on collective encirclement, with personal strengths as assistance. The transformation in this mode is modeled using the capture rate parameter, expressed as δ.

$$\delta = \left(1 - \frac{3 \times EFs}{2 \times MaxEFs}\right)^{\frac{3 \times EFs}{2 \times MaxEFs}} \quad (4)$$

where *EFs* and *MaxEFs* indicate the current number and maximum number of estimates, respectively.

#### a) Individual fishing(when EFs/MaxEFs $<$0.5)

Fishermen disturb the water to float the fish, determine the position of the fish and adjust the direction of exploration. The update formula is as follows:

$$Exp = \frac{f_i - f_{pos}}{f_{max} - f_{min}} \quad (5)$$

$$R = Dis \times \sqrt{|Exp|} \times \left(1 - \frac{EFs}{MaxEFs}\right) \quad (6)$$

$$F_{i,j}^{T+1} = F_{i,j}^T + \left(F_{pos,j}^T - F_{i,j}^T\right) \times Exp + \text{rand} \times s \times R \quad (7)$$

In the formula mentioned above, *Exp* represents the empirical analysis value obtained by the $i$-th fisherman using any other fisherman $p$ (where $pos = 1,2 \cdots$ or $N$, $p \neq i$) as the reference object, with values ranging from -1 to 1.$f_{max}$ and $f_{min}$ represent the lowest and highest fitness values, respectively, following the Tth complete position update. $T$ is the number of iterations fishermen's positions. $F_{i,j}^T$ and $F_{i,j}^{T+1}$ are position of the $i$th fisherman in $j$-dimension after the iterations of $T$th and $(T+1)$th. *Dis* denotes the Euclidean distance between the $i$-th individual and the reference point, while $s$ is a random unit vector in $d$ dimensions.

#### b) group fishing (when EFs/MaxEFs $\geq$ 0.5)

Fishermen utilize nets to enhance their fishing efficiency and collaborate with each other. They organize into random groups of 3-4 members to collectively encircle potential targets. By leveraging their individual mobility, they can

explore the area more comprehensively and accurately. The corresponding formula are outlined below:

$$Centre_c = mean(F_c^T) \tag{8}$$

$$F_{c,i,j}^{T+1} = F_{c,i,j}^T + r_2 \times \left(Centre_c - F_{c,i,j}^T\right) + \left(1 - \frac{2 \times EFs}{MaxEFs}\right)^2 \times r_3 \tag{9}$$

Where $c$ represents a cluster of 3 to 4 individuals whose positions remain unaltered. $Centre_c$ is the target point for group $c$'s encirclement. $F_{c,i,j}^{T+1}$ and $F_{c,i,j}^T$ are the position of the $i^{th}$ fisherman in group $c$ in the $j$-dimension after the $(T+1)^{th}$ and $T^{th}$ updates. $r_2$ represents the speed at which a fisherman moves toward the center, varying individually and falling within the range of $(0,1)$. $r_3$ is the offset of the move, ranging from $(-1, 1)$, and decreases progressively as EFs increase.

*3) Collective capture (exploitation phase)*

All fishermen searched under a uniform strategy, purposefully bringing hidden fish to the same location and around. The position of the fishermen during the trapping process is updated as follows:

$$\sigma = \sqrt{\frac{2\left(1 - \frac{EFs}{MaxEFs}\right)}{\left(\left(1 - \frac{EFs}{MaxEFs}\right)^2 + 1\right)}} \tag{10}$$

$$F_i^{T+1} = Gbest + GD\left(0, \frac{r_4 \times \sigma \times |\, mean(F) - Gbest\,|}{3}\right) \tag{11}$$

Within this group, GD is a Gaussian distribution function with a mean $\mu$ of 0, and its overall variance $\sigma$ decreases from 1 to 0 as the number of evaluations increases. The position of the $i^{th}$ fisherman after the $(T+1)^{th}$ update. *Mean* $(F)$ signifies the matrix of mean values for each dimension at the center of the fishermen's positions, while *Gbest* indicates the global optimum.

## III. PROPOSED ALGORITHM

### A. Adaptive Gaussian Perturbation (AGP)

Adaptive Gaussian Perturbation dynamically enables the optimization algorithm to flexibly balance exploration and exploitation in different iteration stages as follows:

$$\sigma_p = (1 - \frac{EFs}{MaxEFs}) \cdot std(F_i^T) \cdot \exp\left(-\frac{EFs}{MaxEFs/10}\right) \tag{12}$$

$$F_i^{T+1} = F_i^T + \mathcal{N}(0, \sigma_p^2) \tag{13}$$

where $\sigma_p$ is the perturbation strength of the current iteration, The std is the standard deviation of the current solution, $\mathcal{N}(0, \sigma_p^2)$ is a normally distributed random variable with mean 0 and variance $\sigma_p^2$.

### B. Personalized Fishing Strategy (PFS)

PFS is a widely adopted and effective strategy that enhances an algorithm's optimization capability within the search space. Overall, PFS enhances the exploitation capability and accelerates the convergence speed of metaheuristic algorithms (MAs). The principle of PFS is to generate random actions α and β based on the original action step, update the position according to different movements, and change according to the capture parameter δ, making the whole fishing process more closely related.

$$F_i^{T+1} = F_i^T + step \cdot kd \tag{14}$$

$$kd = 0.5 + 0.5 * rand \tag{15}$$

where step represents action choice and kd indicates the action skill proficiency of fishermen, whose value is [0.1,1]. The personalized fishing strategy includes two factors, "freehand capture" and "using tools", which are used to improve the global search capability of the algorithm. The updated formula is shown as follows:

$$step = \begin{cases} \alpha = 2 * \left(1 - \frac{EFs}{MaxEFs}\right) * (0.4 * \delta) * rand > 0.5 \\ \beta = 1 * \left(1 - \frac{EFs}{MaxEFs}\right) \ otherwise \end{cases} \tag{16}$$

### C. Details of ICFOA

CFOA is widely used and easy to implement for optimization tasks. However, its search capabilities (exploration and exploitation) are limited when tackling complex problems, making convergence within a finite number of iterations difficult or even unattainable. To address these challenges, this paper integrates PFS to enhance ICFOA for global optimization. The PFS improves exploitation efficiency and accelerates the convergence rate of the conventional CFOA.

The pseudo-code of ICFOA is shown in Algorithm 1, and Fig. 1 illustrates the flowchart of ICFOA.

| **Algorithm 1** the pseudo-code of ICFOA |
| --- |
| 1.  Initialization parameters |
| 2.  Initialize the population Fisher |
| 3.  **While** (EFs ≤ *MaxEFs*) |
| 4.      Reckon the values of *fit* and get the globally optimal solution(Gbest) |
| 5.      **if** EFs/MaxEFs＜0.5 |
| 6.      Reckon the values of δ by Eq. (4) |
| 7.      Randomly shuffle the order of each fisherman |
| 8.      **If** p＜δ |
| 9.          Using Eq. (7) to update new position of fisher |
| 10.    **Else** |
| 11.        Randomly group the fisherman |
| 12.        Using Eq. (9) to update new position of fisher |
| 13.    **End** |
| 14.    **Else** |
| 15.      Using Eq. (13) to update new position of fisher |
| 16.    **End** |
| 17.**End** |
| 18. Output the globally optimal solution |

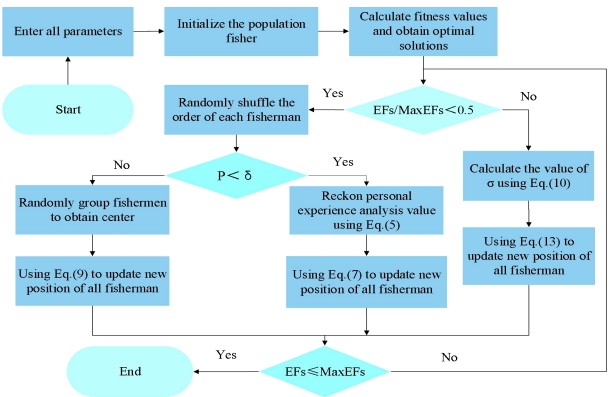

Fig. 1. Flowchart of the proposed algorithm ICFOA

### D. Computation Complexity of ICFOA

Initialization and position updates are the core components of ICFOA. The computational complexity of initialization is O($N \times Dim$), where $N$ represents the population size and $Dim$ denotes the dimensionality. The complexity of fishing with bare hands and using fishing nets varies and can reach up to O($T \times N \times Dim$), where $T$ is the maximum number of evaluations. Both the PFS and AGP methods have a complexity of O($T \times N \times Dim$). Therefore, the overall computational complexity of ICFOA is O($(4 \times T+1) \times N \times Dim$).

## IV. RESULTS OF GLOBAL OPTIMIZATION EXPERIMENTS

This section uses 10 benchmark functions in CEC2020 [7] to evaluate the optimized performance of the proposed working ICFOA. First, the definitions of the 10 benchmark test functions are introduced. Second, the experimental setup and the comparison groups are described in detail, including other well-known MAs.

### A. Definition of 10 Benchmark Functions

Benchmark functions are essential for assessing the performance of various algorithms. This paper selects 10 representative benchmark functions, categorized into: (1) unimodal functions (F1-F4) and (2) multimodal functions (F5-F10). Table 1 provides a detailed description of these functions, and D denotes the dimensionality. Unimodal functions have a single global optimal solution, making them suitable for evaluating the exploitation capability of metaheuristic algorithms (MAs). Conversely, multimodal functions possess multiple local optima and a single global optimum. providing a basis for assessing the exploration capability of MAs and their ability to escape local optima.

TABLE I. DEFINITION OF 10 BENCHMARK FUNCTIONS

| No | Property | D | Range | $f_{min}$ |
|---|---|---|---|---|
| F1 | Unimodal Function | | | 100 |
| F2 | Basic Functions | | | 1100 |
| F3 | | | | 700 |
| F4 | | | | 1900 |
| F5 | Hybrid Functions | 100 | $[-100,100]^D$ | 1700 |
| F6 | | | | 1600 |
| F7 | | | | 2100 |
| F8 | Composition Functions | | | 2000 |
| F9 | | | | 2400 |
| F10 | | | | 2500 |

### B. Experimental Configuration

We utilize aforementioned functions to evaluate the performance of ICFOA. To ensure the experiment's representativeness, we compare the enhanced algorithm with the basic CFOA and five widely used metaheuristic algorithms, including ROA [8], AOA [9], SFO [10], SHO [11], and SCA [12]. To ensure an unbiased comparison, we define the maximum number of evaluations as $T = 100,000$, the group size as $N = 30$, and the dimensionality as $Dim = 10$. Furthermore, each test are conducted independently 30 times, with the best results emphasized in bold.

TABLE II. ALGORITHM PARAMETER SETTINGS

| Algorithm | Parameters |
|---|---|
| ICFOA | α=0.4, |
| CFOA | α=0.4,β=0.5 |
| ROA | c=0.2 |
| AOA | α=5; μ=0.5; |
| SCSO | S=2, R=[-1,1]; |
| SHO | u=0.03, v=0.03, l=0.03 |
| SCA | B=3 |

### C. Statistical Analysis of 10 Benchmark Functions

This part compares ICFOA with five foundational algorithms across 10 benchmark functions, focusing on the optimal value (Best), mean value (Mean), and standard deviation (Std) [13]. Table 3 provides the details of the experimental outcomes. From the table, it is evident that ICFOA performs well on most functions, often achieving the minimum Best, Mean, and Std values. In particular, for F1-F6, ICFOA consistently attains the theoretical optimal solution, whereas CFOA solely approximates it, highlighting ICFOA's superior exploitation capability. For F2, ICFOA demonstrates better global optimization performance than other prominent algorithms. Despite ICFOA finds solely a suboptimal solution for F3, its precision in convergence surpasses that of the others. For F4, F5, and F6, ICFOA reaches the theoretical optimum. However, for F4 and F9, ICFOA's performance is slightly inferior to the SFO algorithm.

Considering that MAs are stochastic algorithms, this paper employs the Wilcoxon rank-sum test to enhance the statistical analysis and assess the significance of the results. Notably, if the p value is below 0.06, there is a significant difference between the two data groups. Conversely, when p value is at least 0.06, this indicates minimal difference between the data sets. Additionally, "NAN" is used in this paper to represent cases where no significant difference exists between the groups. The detailed results of the Wilcoxon rank-sum test are displayed in Table 4. The results show that functions F4, F6, and F10 contain 'NAN,' because the optimization results of ICFOA, CFOA, and SHO all achieve the theoretical optimal solution, leading to minimal differences among the three data groups. For other functions, The performance of ICFOA differs significantly from the other algorithms. However, the Wilcoxon rank-sum test solely measures the statistical difference between algorithms, not their overall performance. Consequently, by integrating the insights from Tables 3 and 4, it is evident that the improvements to CFOA presented in this paper are highly effective.

TABLE III. Statistics about the 10 test functions in the cec2020

| FUNCTIONS | | ICFOA | CFOA | ROA | AOA | SFO | SHO | SCA |
|---|---|---|---|---|---|---|---|---|
| F1 | Best | **123.9163085** | 1303.684774 | 2435.264611 | 2484.316844 | 2898.991125 | 27552.63251 | 17016.33261 |
| | Mean | **4591.728112** | 202423937.5 | 26245.21152 | 26812.31168 | 28990.51251 | 28799.22518 | 18504.45167 |
| | Std | **4137.802543** | 68441680.22 | 1924016987 | 9083474822 | 1924069.487 | 4948400618 | 13440911059 |
| F2 | Best | **12244.11095** | 16605.71202 | 29808.77688 | 28990.18679 | 35461.62181 | 34698.94999 | 31568.08871 |
| | Mean | **18161.78534** | 23564.86621 | 31381.50853 | 30430.72471 | 35488.16642 | 36163.21061 | 32488.08215 |
| | Std | **3545.047924** | 4192.739178 | 114655.1359 | 9077.919527 | 24045.26547 | 103551.2026 | 5910.424614 |
| F3 | Best | **894.5056084** | 1683.533974 | 3908.81377 | 3806.349814 | 4217.387789 | 4066.605567 | 3489.622851 |
| | Mean | **927.3464718** | 2120.641269 | 4011.883834 | 3909.740683 | 4221.83685 | 4204.438691 | 3765.087815 |
| | Std | 40.84472928 | 270.9768306 | 56.03266113 | 53.70544871 | **30.76279062** | 75.21084023 | 180.6559676 |
| F4 | Best | **1911.571228** | 1972.715291 | 1926.112879 | 1913.360194 | 1942.664109 | 1926.352858 | 5871.613809 |
| | Mean | **1903.184835** | 2003.719718 | 1938.664563 | 1911.360491 | 1908.059367 | 1933.542475 | 33306.19437 |
| | Std | **1.974573432** | 20.75907556 | 48.00546352 | 7.62E+00 | 5.236162188 | 10.3265855 | 41611.3348 |
| F5 | Best | 771627.8334 | 2622829.749 | 648036492.7 | 897453834.9 | 1910249217 | 1782936241 | 274232285.4 |
| | Mean | 1572147.454 | 4807902.622 | 1050612701 | 1321917258 | 3048989087 | 2526379882 | 446370493.4 |
| | Std | 631467.6545 | 1843676.178 | 335444387.9 | 299582965.4 | 550649621.7 | 386586151.4 | 120087797.6 |
| F6 | Best | **3770.079006** | 6119.393228 | 20830.74823 | 19888.73262 | 41146.71723 | 33215.10053 | 12668.3214 |
| | Mean | **4385.997062** | 6830.04364 | 25109.30245 | 25960.73394 | 46688.73657 | 40988.94545 | 15436.56211 |
| | Std | 283.7781916 | 495.5286566 | 5039.633742 | 4268.600462 | 1947.267797 | 4555.865169 | 1484.988875 |
| F7 | Best | 548601.3933 | 1444325.148 | 202662857.8 | 192645390.1 | 469380299.5 | 583159047.5 | 78210737.68 |
| | Mean | 1424855.595 | 2559045.072 | 334117181.3 | 402478936.1 | 470079347.8 | 842041986.2 | 142352823.7 |
| | Std | 432943.2661 | 961142.875 | 80378422.23 | 174899077.8 | 5862372.585 | 141415505.3 | 45868952.47 |
| F8 | Best | **15713.23631** | 19186.2209 | 31516.95746 | 31589.84888 | 37352.98584 | 36694.26248 | 32989.46542 |
| | Mean | **21014.047** | 25657.00443 | 33549.03571 | 32922.86116 | 37361.35642 | 38213.17431 | 34579.26347 |
| | Std | **3786.519778** | 4731.682899 | 6231.863806 | 7026.180794 | 5623.056834 | 7575.350727 | 6658.448304 |
| F9 | Best | 3382.896377 | 4089.081127 | 7958.084137 | 9625.401505 | **1353.90174** | 10910.13186 | 6418.969662 |
| | Mean | **3423.654041** | 4254.403066 | 9835.102409 | 11574.95115 | 13419.51066 | 13443.41938 | 6865.220732 |
| | Std | 22.39302112 | 114.1072971 | 1617.806356 | 1166.977396 | 254.2600235 | 1589.567702 | 212.5751176 |
| F10 | Best | **3390.009962** | 3725.611385 | 21898.12822 | 25910.18877 | 35026.3259 | 30898.44744 | 15251.35182 |
| | Mean | **3539.668413** | 3814.755832 | 26384.31184 | 29091.06983 | 35026.85245 | 32999.06763 | 17488.29627 |
| | Std | 36.32133408 | 81.68764222 | 2479.008781 | 1985.004222 | 52.42536743 | 890.9886491 | 1429.3475 |

## D. Convergence Analysis of ICFOA and Comparative Algorithms

The convergence curve is a crucial metric for assessing an algorithm's performance. Figure 2 displays the convergence curves of these algorithms on several benchmark functions. The results indicate that ICFOA demonstrates a notably faster convergence speed on the unimodal functions F1 and F2. For F5, although ICFOA does not achieve the best performance, its results are still very close to the optimal solution, underscoring ICFOA's strong exploitation capability. In the cases of functions F6 and F7, ICFOA performs similarly to the ROA algorithm but with a quicker convergence rate. For F9, the ICFOA, CFOA, and SCA algorithms all converge rapidly. Overall, it is evident that ICFOA exhibits excellent convergence capabilities across different types of functions.

TABLE IV. Statistical Results of Algorithms on 10 Benchmark Functions using Wilcoxon Rank-sum Test

| Function | ICFOA vs. | | | | | | |
|---|---|---|---|---|---|---|---|
| | CFOA | ROA | AOA | SCA | SFO | SHO | SCA |
| F1 | 3.34×10⁻⁰¹ | 1.21×10⁻¹² | 1.21×10⁻¹² | 1.21×10⁻¹² | 1.21×10⁻¹² | 1.21×10⁻¹² | 1.21×10⁻¹² |
| F2 | 1.21×10⁻¹² | 1.21×10⁻¹² | 1.21×10⁻¹² | 1.21×10⁻¹² | 1.21×10⁻¹² | 1.21×10⁻¹² | 1.21×10⁻¹² |
| F3 | 4.57×10⁻¹² | 1.21×10⁻¹² | 1.21×10⁻¹² | 1.21×10⁻¹² | 1.21×10⁻¹² | 1.21×10⁻¹² | 1.21×10⁻¹² |
| F4 | NaN | 1.21×10⁻¹² | 1.21×10⁻¹² | NaN | 1.21×10⁻¹² | 1.21×10⁻¹² | 1.21×10⁻¹² |
| F5 | 4.83×10⁻⁰¹ | 1.07×10⁻⁰⁷ | 3.02×10⁻¹¹ | 3.02×10⁻¹¹ | 1.21×10⁻¹² | 1.21×10⁻¹² | 1.21×10⁻¹² |
| F6 | 4.62×10⁻¹⁰ | 3.02×10⁻¹¹ | 3.02×10⁻¹¹ | 3.02×10⁻¹¹ | 1.21×10⁻¹² | NaN | 1.21×10⁻¹² |
| F7 | 6.10×10⁻⁰³ | 3.11×10⁻⁰¹ | 7.98×10⁻⁰² | 3.02×10⁻¹¹ | 1.21×10⁻¹² | 1.21×10⁻¹² | 1.21×10⁻¹² |
| F8 | 1.07×10⁻⁰⁷ | 3.02×10⁻¹¹ | 1.22×10⁻⁰² | 3.02×10⁻¹¹ | 1.21×10⁻¹² | 1.21×10⁻¹² | 1.21×10⁻¹² |
| F9 | 1.21×10⁻¹² | 4.57×10⁻¹² | 1.22×10⁻¹² | 1.21×10⁻¹² | 1.21×10⁻¹² | 1.21×10⁻¹² | 1.21×10⁻¹² |
| F10 | NaN | 1.21×10⁻¹² | NaN | 1.21×10⁻¹² | 1.21×10⁻¹² | 1.21×10⁻¹² | 1.21×10⁻¹² |

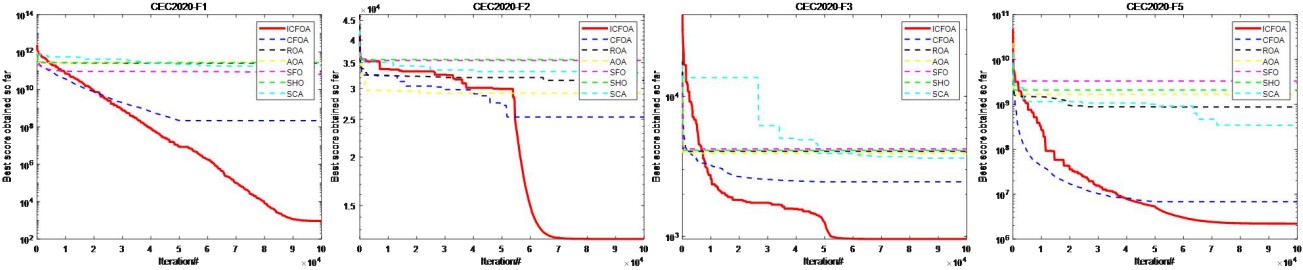

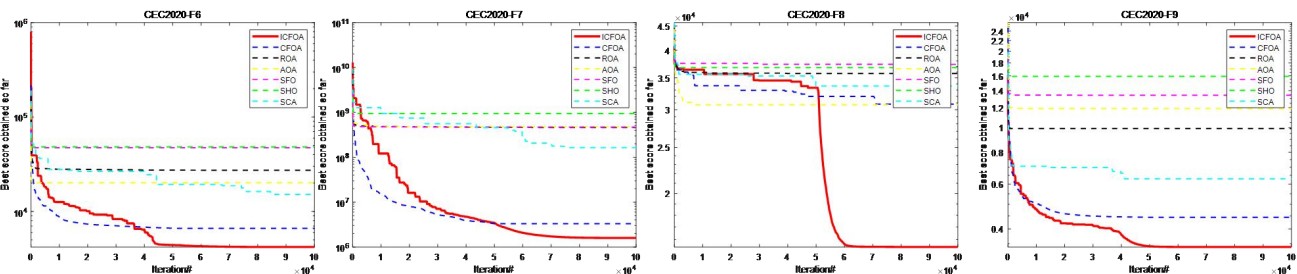

Fig. 2. Convergence of ICFOA and Comparison Algorithm on Some Functions

## V.    REDUCER STRUCTURE MODEL DESIGN PROBLEM

The model for the reducer design problem is illustrated in Figure 3. The primary objective is to minimize the mass of the reducer while satisfying the given constraints. This problem involves seven decision variables and eleven constraints. For detailed descriptions of the variables x1 to x11), refer to reference [14]. The mathematical formulation of the problem is identical to that in reference [14].

Table 5 presents the comparison results of ICFOA, CFOA, ROA, AOA, SFO, SHO, and SCA in solving the reducer design problem. It's evident that ICFOA produces strong results while effectively meeting the constraints.

TABLE V. COMPARISON RESULTS OF DIFFERENT OPTIMIZATION ALGORITHMS

| Algorithm | ICFOA | CFOA | ROA | AOA | SFO | SHO |
|---|---|---|---|---|---|---|
| $x_1$ | **0.501** | 0.531 | 0.54 | 0.57 | 0.568 | 0.525 |
| $x_2$ | **1.231** | 1.262 | 1.257 | 1.27 | 1.241 | 1.239 |
| $x_3$ | **0.515** | 0.540 | 0.563 | 0.54 | 0.517 | 0.528 |
| $x_4$ | **1.096** | 1.149 | 1.167 | 1.14 | 1.246 | 1.200 |
| $x_5$ | **0.517** | 0.558 | 0.631 | 0.64 | 0.534 | 0.781 |
| $x_6$ | **0.486** | 0.511 | 0.538 | 0.54 | 0.941 | 1.160 |
| $x_7$ | **0.503** | 0.510 | 0.528 | 0.50 | 0.525 | 0.564 |
| $x_8$ | 0.346 | 0.351 | 0.472 | **0.27** | 0.332 | 0.340 |
| $x_9$ | **0.342** | 0.344 | 0.356 | 0.36 | 0.336 | 0.319 |
| cost | **23.01** | 23.20 | 23.38 | 23.55 | 23.45 | 23.92 |

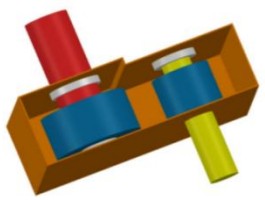

Fig. 3. Reducer structure model

## VI.    CONCLUSION

Building on CFOA, this paper introduces a strategy called Personalized Fishing Strategy (PFS) to propose an improved CFOA (ICFOA). Although CFOA has been applied to solve various design problems, ICFOA addresses some of its limitations, for example, a tendency to become trapped in algorithm stagnation and local optima. By incorporating PFS, ICFOA enhances global search capability, thereby improving its ability to escape local optima. To assess the effectiveness of ICFOA, this paper compared it with five other well-established algorithms using 10 benchmark functions. The results demonstrate that ICFOA performs exceptionally well across most benchmark functions and practical engineering problems.

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
