# OpenReview forum: "Improved Catch Fish Optimization Algorithm with Personalized Fishing Strategy for Global Optimization"
_IEEE.org/ICIST/2024/Conference — IEEE ICIST 2024 Conference Submission_

### Official Review · Reviewer_PVhK · 2024-08-21
**accept**

**Rating:** 7
**Confidence:** 3

**Review:**

Comment: This paper proposes animproved catch fish optimization algorithm (ICFOA) based on personalized fishing strategies. Based on the CFOA, a strategy called Personalized Fishing Strategy (PFS) was added. The theory is correct and can be accepted after responding the following comments.
(1) More comprehensive literature review is needed to clarify the research gap and research motivation.
(2) There are some typos and grammar errors. The authors should have a native English speaker or software packages to perform the editing check.
(3) In the end of the conclusions, some research directions are suggested to be added.

---

### Official Review · Reviewer_eSs4 · 2024-08-23
**this work is well organized and appears potentially interesting, it can be accepted with a little modification.**

**Rating:** 7
**Confidence:** 3

**Review:**

The main work of the paper involves the development and implementation of an improved algorithm, termed ICFOA, which incorporates personalized fishing strategies and adaptive Gaussian perturbation to overcome the limitations of CFOA, particularly its tendency to fall into local optima and its low convergence rate. In general, this work is well organized and appears potentially interesting, it can be accepted with a little modification.
1.	What makes this system innovative compared to others?
2.	What are the future research directions discussed in the article?
3.	Please analyze the simulation results in detail.
4.	There are a few typos in this paper which should be corrected. And there are some notions missed. Please make some corrections.

---

### Official Review · Reviewer_D2fL · 2024-08-23
**Catch Fish Optimization Algorithm (CFOA) is a new meta-heuristic optimization algorithm with human behavior. In this algorithm, search agents simulate the process of rural fishermen fishing in the pond. Therefore, the CFOA generally consists of two phases of the update: the exploration phase and the exploitation phase. However, it still falls under the local optimum and has a low convergence rate. To this end, we propose an improved catch fish optimization algorithm(ICFOA) based on personalized fishing strategies. First, the adaptive Gaussian perturbation is adopted to the exploration stage process to increase the global search capability, expand the search range, and improve efficiency while avoiding falling into the local optima. Then, based on the personalized fishing strategy, the personal position of fishermen is updated by randomly selecting "freehand fishing" factors or "using fishing net" factors to accelerate the algorithm's convergence speed. Furthermore, comparative experiments were performed using the CEC2020 test suite to compare the performance of ICFOA and other excellent meta- heuristics. Finally, Wilcoxon's rank-sum test was used to verify the validity of our statistical experimental results. Moreover, the performance of ICFOA in reducer design also indicates that ICFOA can get the optimal solution in solving practical engineering optimization problems. The results show that ICFOA has a more competitive performance than the original CFOA. Comments for this submission are given as follows.**

**Rating:** 7
**Confidence:** 3

**Review:**

(1)The formatting of the formulas in this paper is more complete, but should the two equations in equation (16) be aligned to the left. The authors should double-check the full text.
(2)The language format of this paper is relatively standard, but whether there are places where no attention has been paid to the issue of spaces, such as “ROA [8], AOA[9], SFO[11]” in “B. Experimental Setup”, “ROA [8]” with a space and “AOA[9], SFO[11]” without, the authors are requested to standardise. In addition, the authors should double-check the whole paper.
(3)The literature format of this paper is not uniform, so the authors are requested to check and correct the attitude, the same literature format, and make changes carefully.

---

### Decision · Program_Chairs · 2024-09-06

Accept (Oral)